# Identification and Characterization of a Novel *CCDC6::CASP7* Gene Rearrangement in an Advanced Colorectal Cancer Patient: A Case Report

**DOI:** 10.3390/ijms252312665

**Published:** 2024-11-26

**Authors:** Juan Carlos Montero, Raquel Tur, Andrea Jiménez-Perez, Elena Filipovich, Susana Alcaraz, Marta Rodríguez, Mar Abad, José María Sayagués

**Affiliations:** 1Department of Pathology, Institute for Biomedical Research of Salamanca (IBSAL), University Hospital of Salamanca, Paseo de San Vicente, 58-182, 37007 Salamanca, Spain; jcmon@usal.es (J.C.M.); abjimenez@saludcastillayleon.es (A.J.-P.); smalcaraz@saludcastillayleon.es (S.A.); martarodriguez@saludcastillayleon.es (M.R.); marabad@usal.es (M.A.); 2Centro de Investigación Biomédica en Red Cáncer (CIBERONC), 28029 Madrid, Spain; 3Department of Pathology, University Hospital of Avila, 05004 Avila, Spain; rtur@saludcastillayleon.es; 4Department of Oncology, University Hospital of Avila, 05004 Avila, Spain; efilipovich@saludcastillayleon.es

**Keywords:** advanced colorectal cancer, NGS, *CCDC6::CASP7*, rearrangements

## Abstract

Despite the existence of effective therapy options for patients with localized colorectal cancer, advanced-stage patients have limited therapies. Genomic profiling is a promising tool for guiding treatment selection as well as patient monitoring. Here, we describe a novel gene rearrangement (*CCDC6::CASP7*) detected in a patient with advanced colorectal cancer that could be a therapeutic target. The patient underwent surgical resection but died after the operation from fecal peritonitis. To our knowledge, this is the first report in which the *CCDC6::CASP7* gene rearrangement has been described in an advanced colorectal adenocarcinoma patient.

## 1. Introduction

Worldwide, sporadic colorectal cancer (CRC) is the second most common cause of cancer-related deaths and the third most frequently malignancy diagnosed [1]. Up to a quarter of CRC cases present with stage IV disease, and half of the cases diagnosed at an early stage go on to develop metastases over the course of the disease. Early detection and effective treatment strategies are crucial to improving survival rates and reducing the burden of this malignancy. Patients with advanced disease have few treatment options, which contributes significantly to mortality [2]. Thus, the five-year survival rate for advanced colorectal cancer (aCRC) is <10% [3]. Precision medicine for aCRC has shown that as our understanding of the genomic landscape of tumors deepens, the therapeutic paradigm expands. In this regard, the discovery of new biomarkers and their integration into clinical practice has enhanced the ability to diagnose, predict prognosis and select more effective treatments. However, chromosomal rearrangements that are potentially characteristic of such advanced stages of the disease are complex and, so far, only partially understood. In the present study, we found that only 1 of the 73 consecutive patients with aCRC analyzed by next-generation sequencing (NGS) techniques showed a *CCDC6::CASP7 (coiled-coil domain-containing 6::caspase 7)* rearrangement that could be a therapeutic target. In fact, low levels of CCDC6 expression in non-small-cell lung (NSCLC) cancer have been shown to confer resistance to chemotherapy and sensitivity to the poliADP-ribosa polimerasa (PARP) inhibitor [4]. Moreover, a synergistic effect on NSCLC cells lines was observed when the inhibitor was combined with cisplatin [4]. The same thing happens in malignant pleural mesothelioma [5]. Morra et al. [6] also demonstrated in the HCT116 cell line of CRC that low CCDC6 expression increases sensitivity to olaparib [6]. However, it is still unclear whether this rearrangement itself affects the altered expression or function of the caspase 7 protein.

In this article, we present the case of an 87-year-old male with aCRC carrying *CCDC6::CASP7* rearrangements, caused by an inversion of the intergenic segment between the two genes. In addition to fusion, somatic mutations of proto-oncogenes *PIK3CA (phosphoinositide-3-kinase catalytic subunit alpha)* (*E542K*) and *KRAS (kirsten rat sarcoma viral oncogene homolog)* (G12C) were detected (Appendix A). The patient underwent surgical resection but died from fecal peritonitis after the operation.

## 2. Case Presentation

An 87-year-old man was referred by his physician for abdominal pain of months of evolution and recent vomiting. Computed tomography (CT) imaging showed a perforated cecum neoplasm with hepatic metastases. Immediate right hemicolectomy was performed in May 2019, in the Surgery Department of Hospital Nuestra Señora de Sonsoles (Ávila, Spain). During surgery, an enlarged tumor was observed that reached the wall of the right iliac fossa and featured abundant purulent and fecaloid liquid in the abdominal cavity. The pathology department received a 26 cm segment of the right colon with a perforated area. Pathological findings revealed a moderately differentiated adenocarcinoma with microsatellite stability (MSS). It was microscopically confirmed that the tumor infiltrates serosa (pT4a) with metastasis in 7 out of 16 lymph nodes analyzed (pN2b).

The Oncomine Precision Assay (OPA), which enables the simultaneous detection of hotspot mutations (substitutions, insertions, and deletions), copy number variations (CNVs) and gene fusions across 50 cancer-related genes using the Ion Torrent GX5 Chip, was used to perform NGS. Using the Oncomine^TM^ Reporter program (Thermo Fisher Scientific; Waltham, MA, USA), data were analyzed automatically. The Variant Matrix Summary filter (version 5.16) was used to filter the findings. This filter offers an overview of hotspot mutations, including the modifications to the amino acids and the allelic frequency. Furthermore, it indicates the number of readings of detected fusions and provides numerical values for CNVs. The patient’s genomic profile revealed oncogenic mutations in *PIK3CA* (E542K; allele frequency: 17.8%) and *KRAS* (G12C; allele frequency: 33.1%) proto-oncogenes, as well as the *CCDC6::CASP7* rearrangement (Figure 1 Panel A, and Appendix A). The NGS results show that the rearrangement produces two different transcripts. In the most frequent transcript (70% of reads), the first 95 amino acids of exon 1 of *CCDC6* are fused with the first two amino acids of exon 2 of *CASP7*. In contrast, the second transcript (30% of reads) preserves the first 82 amino acids of exon 1 of *CCDC6*, and due to a deletion of a base, a frameshift occurs that allows exon 2 of *CASP7* to be read correctly (Figure 1; panel A). 

In addition, to determine whether the rearrangement is caused by a deletion or inversion of the intergenic segment between the genes *CCDC6* (10q21.2) and *CASP7* (10q25.3), we performed fluorescence in situ hybridization (FISH) to assess the loss of the *PTEN* (*phosphatase and tensin homolog*) gene, located at 10q23.3. FISH analysis revealed a normal number of hybridization signals for both the gene and the centromere of chromosome 10, suggesting that the fusion is caused by an inversion of the intergenic segment between the two genes (Figure 1; panel B).

Informed consent was obtained from the patient prior to their participation in the study, in accordance with the Declaration of Helsinki. The study was approved by the local ethics committee at the University Hospital of Salamanca (Salamanca, Spain): code 2024/07; year, 2024.

## 3. Discussion

Finding new treatments for cancer is a medical necessity, particularly when the disease is advanced. In this scenario, aCRC provokes a higher death rate, primarily because the antitumoral drugs that are currently on the market fail. To improve the prognosis of cancer patients, it is crucial to identify novel genetic changes in tumor samples that are amenable to pharmacological treatment. NGS techniques have revolutionized molecular tumor testing in oncology by offering the simultaneous assessment of many gene regions using formalin-fixed, paraffin-embedded clinical samples. NGS is currently used in pathological anatomy services, primarily in NSCLC, to identify molecular changes that can be treated with medication. To find novel, potentially treatable molecular changes in a subset of 73 consecutive aCRC patients whose treatment choices are scarce, we therefore undertook an NGS investigation.

In this case report, we describe for the first time a rearrangement of *CCDC6::CASP7* in aCRC. *CCDC6* was first identified in papillary thyroid carcinoma as a result of the rearrangement of an unknown amino-terminal sequence with the tyrosine kinase domain of the *RET* proto-oncogene [7]. In addition, *CCDC6* has subsequently been described as being involved in other rearrangements, in addition to *RET*, in different neoplasias [8]. In In fact, up to 26 different rearrangements involving the *CCDC6* gene have been noted (FusionGDB 2.0 database). Several studies have shown that *CCDC6* can fuse with *RET* in CRC [9,10,11,12]. This fusion has been associated with aggressive tumor behavior and poor prognosis in this type of neoplasia [9]. In general, these types of fusions lead to increased kinase activity and result in the silencing or disruption of *CCDC6* normal function. 

Here, we identify the mechanism leading to the generation of the oncogenic sequence *CCDC6::CASP7* by NGS and FISH techniques. FISH analysis revealed that the fusion is caused by an inversion of the intergenic segment between the two genes. These findings are in line with the rearrangements found by Pierotti et al. [13] on the long arm of chromosome 10, juxtaposing the tyrosine kinase domain of the *RET* gene with the 5′ end of the locus *D10S170*. Similarly, Hamatani et al. [14] demonstrated that *ACBD5::RET* fusions are produced by pericentric inversion, inv(10)(p12.1;q11.2), in papillary thyroid cancer. 

CCDC6 is a protein that is ubiquitously expressed and involved in several critical cellular processes, including transcription regulation, DNA damage response (DDR), and apoptosis [4,8,15,16]. Truncated forms of *CCDC6*, such as those retaining only the first 101 amino acids (commonly found in *RET* fusions), act as dominant negatives, impairing the protein’s pro-apoptotic function and proper nuclear localization [17]. On the other hand, caspase 7 is a protein involved in apoptosis (programmed cell death), and its dysfunction or genetic alterations, such as fusions, can disrupt this process [18]. *CASP7* gene fusions could produce abnormal proteins that interfere with the ability of cells to effectively execute apoptosis, potentially contributing to tumor cell survival and cancer progression.

In the fusion that we have identified, the majority transcript should produce a chimeric protein in which the function of both proteins is lost. The same occurs with the minor transcript, which, despite expressing exon 2 of caspase 7, lacks exon 1, which is critical for the complete structure and proper function of the protein. Therefore, this transcript would also result in a loss of function for both proteins. 

The patient’s mutational status revealed that, in addition to the *CCDC6::CASP7* fusion, he had two mutations, one in *KRAS* and one in *PIK3CA*, which ruled him out as a candidate for targeted therapy with anti-EGFR (epidermal growth factor receptor) antibodies. However, a study conducted in colorectal cancer demonstrated that the *CCDC6::RET* fusion is mutually exclusive with other important driver mutations, such as *KRAS*, *BRAF* and *PIK3CA* [12]. It has been reported that the loss or inactivation of *CCDC6* leads to premature mitotic entry, reduces the DDR and consequently prevents the induction of apoptosis [4,16]. These findings suggest that tumors with *CDCC6* loss of function could be sensitive to a combination of antimitotic drugs and compounds that affect DNA damage [19,20]. In fact, previous studies indicated that the loss of function of the *CCDC6* gene confers resistance to cisplatin and sensitizes tumor cells to PARP inhibitors [4,21]. 

Mutations in the *KRAS* gene often drive oncogenic signaling, which increases genomic instability and oxidative stress [22]. This might indirectly impact the *CCDC6* gene, as its role in DDR becomes more critical in cells with high mutation loads [16]. The loss or dysfunction of *CCDC6* could exacerbate the oncogenic effects of *KRAS*, as impaired DDR would allow for the unchecked accumulation of mutations, fueling cancer progression. In addition, if *CCDC6* is mutated or downregulated, the lack of genomic stability might indirectly enhance *KRAS*-driven oncogenesis by promoting a permissive environment for tumor growth. In this regard, DDR inhibitors (like PARP inhibitors) in combination with inhibitors of the MAPK/ERK pathway may exploit the vulnerabilities created by *CCDC6* loss and *KRAS* mutation [23].

In summary, we provide comprehensive evidence that *CCDC6* plays a pivotal role in maintaining genomic stability by modulating the DDR and cell cycle progression. Its loss of function or inactivation, particularly through chromosomal translocation with *CASP7*, enhances susceptibility to cancer by impairing accurate DNA repair and promoting premature cell division, in the absence of cell death. The identification of *CCDC6:CASP7* rearrangements in patients with aCRC could help select patients who might benefit from treatment with PARP inhibitors alone or in combination with other treatments. In this context, techniques such as FISH could enable the precise detection of these gene rearrangements and help determine their true incidence in aCRC, as well as in other neoplasms. Additional prospective studies in larger series of patients are required to confirm the clinical utility of this biomarker.

To our knowledge, this is the first report identifying the *CCDC6::CASP7* rearrangement in a patient with aCRC, making it a target for further research and potential therapeutic intervention. 

## Figures and Tables

**Figure 1 ijms-25-12665-f001:**
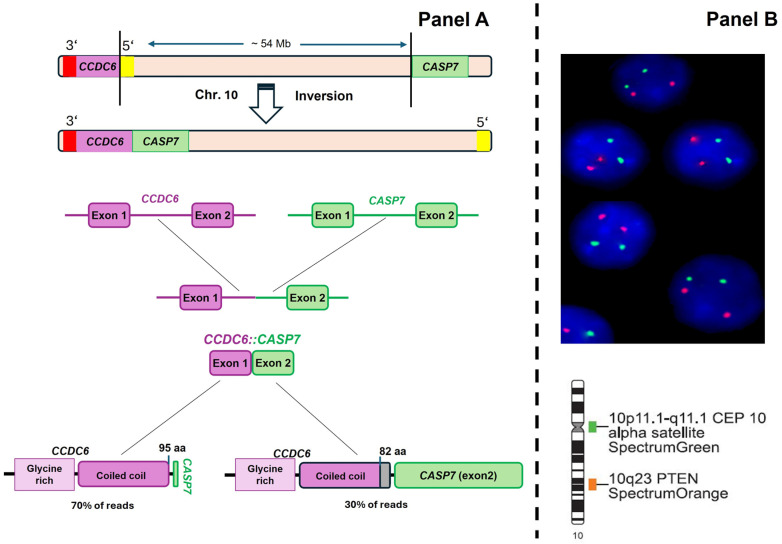
Molecular characterization of the *CCDC6::CASP7* rearrangement detected in a patient with aCRC by NGS. Panel (**A**): In the aCRC specimen, a chromosomal inversion in chromosome 10 juxtaposes the 5′ end of exon 1 of the *CCDC6* gene with the 3′ end of exon 2 of the *CASP7* gene, resulting in the fusion oncogene *CCDC6::CASP7*. The NGS results show that the rearrangement produces two different transcripts: (i) the first 95 amino acids of exon 1 of *CCDC6* are fused with the first two amino acids of exon 2 of *CASP7* (70% of reads), and (ii) the second transcript preserves the first 82 amino acids of exon 1 of CCDC6, and due to a deletion of a base, a frameshift occurs that allows exon 2 of *CASP7* to be read correctly (30% of reads). Panel (**B**): Representative pictures of nuclei counterstained with DAPI (blue) and hybridized with a probe for the centromere of chromosome 10 (green spots) and a probe for the *PTEN* gene (red spots). All the nuclei analyzed showed a normal number of hybridization signals: two for the centromere of chromosome 10 and two for the *PTEN* gene.

## Data Availability

Data is contained within the article and Appendix A.

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
