# Peer review of "Identification and Characterization of a Novel CCDC6::CASP7 Gene Rearrangement in an Advanced Colorectal Cancer Patient: A Case Report"

_ijms, 2024, doi:10.3390/ijms252312665_

Round 1
Reviewer 1 Report
Comments and Suggestions for Authors Authors report a new gene fusion CCDC6-CASP7 in a colorectal cancer patient using a targeted NGS panel. They perform PTEN and chr 10 centromere FISH and find no irregularities, concluding that the fusion must have arisen from an inversion. Authors then speculate on the broader clinical relevance. However, no secondary validation is provided such as PCR or RTPCR which is required, since NGS is not 100% reliable for calling fusions. The FISH data does not validate the fusion, it only tests for chromosome 10 integrity. No proposed functional impact of the fusion is provided, authors should examine the functionality of the truncated domains. No functional validation is provided; in order to prove oncogenicity it should be expressed in a model system. In short, this is an n=1 study with no evidence that the fusion is of any clinical or therapeutic interest.Author Response
REVIEWER COMMENT 1: Authors report a new gene fusion CCDC6-CASP7 in a colorectal cancer patient using a targeted NGS panel. They perform PTEN and chr 10 centromere FISH and find no irregularities, concluding that the fusion must have arisen from an inversion. Authors then speculate on the broader clinical relevance. However, no secondary validation is provided such as PCR or RTPCR which is required, since NGS is not 100% reliable for calling fusions.
AUTHOR RESPONSE 1: We understand the reviewer's concern. Unfortunately, there are no commercial probes available for validating this fusion using FISH techniques. On the other hand, validation of the fusion using RT-PCR is difficult to perform due to the high fragmentation/degradation of RNA present in the paraffin-embedded sample. However, the number of reads detected in the fusion during the NGS experiments leaves no doubt about the presence of this new rearrangement.
REVIEWER COMMENT 2: No proposed functional impact of the fusion is provided, authors should examine the functionality of the truncated domains. No functional validation is provided; in order to prove oncogenicity it should be expressed in a model system.
AUTHOR RESPONSE 2: We understand the importance and relevance of the reviewer's suggestion regarding the possible functional impact of the fusion. Conducting a functional analysis, such as expressing this fusion in cell lines to observe its impact on proliferation or protection from apoptosis, would require a significant amount of additional time and, in our opinion, is not necessary for this case report. However, in the discussion section, we speculate on its possible biological role because other papers have already described the role of similar fusions.

Reviewer 2 Report
Comments and Suggestions for Authors
The authors of this paper describe CCDC6::CASP7 gene rearrangement as a novel mutation detected in a colorectal cancer patient. The report of a novel mutation in cancer patients is very important, however I consider this report to be a little scarce in details.
I state below some of the propositions to improve the manuscript and describe in more detail this case report.
1. Is this variant mentioned in any clinical database? If not, did the authors consider reporting it in ClinVar or any similar database so it can be available and possibly confirmed by other authors? This and similar databases can help verify by different investigators is the variant more or less probably pathogenic which makes it very important for the possible use as a biomarker for directing the treatment.
2. Did the authors ask for a written consent for the publishing of this case? Written consent from the patient or his next of kin. Please state clearly in the manuscript.
3. Was this investigation approved by the ethics committee? If so, please state clearly in the manuscript and provide details of the ethics committee approbation (code and year).
4. Was there any other analysis done to the patient? HER2? KI67? Even if it is not directly connected to this variant, it still could be of interest for the case analysis.
5. Does the patient have a family pedigree with known medical history? Are there any relatives with tumor in the family history? If so, could the authors consider adding it to the manuscript.
6. This manuscript should be revised by the native English speaker as there are some minor errors throughout the manuscript.
Author Response
REVIEWER COMMENT 1: The authors of this paper describe CCDC6::CASP7 gene rearrangement as a novel mutation detected in a colorectal cancer patient. The report of a novel mutation in cancer patients is very important, however I consider this report to be a little scarce in details.
AUTHOR RESPONSE 1: We would like to thank the reviewer for highlighting the importance of detecting a new fusion in a patient with colorectal cancer.
REVIEWER COMMENT 2: Is this variant mentioned in any clinical database? If not, did the authors consider reporting it in ClinVar or any similar database so it can be available and possibly confirmed by other authors? This and similar databases can help verify by different investigators is the variant more or less probably pathogenic which makes it very important for the possible use as a biomarker for directing the treatment.
AUTHOR RESPONSE 2: We thank the reviewer for this suggestion. This fusion has not yet been mentioned or reported in databases such as ClinVar, FusionGDB2, or the Mitelman Database. Our intention is to register this new fusion in one of the aforementioned databases. As the reviewer mentions, its inclusion in the database will help to verify the degree of pathogenicity and therefore determine whether it can be used as a biomarker in future targeted treatments.
REVIEWER COMMENT 3: Did the authors ask for a written consent for the publishing of this case? Written consent from the patient or his next of kin. Please state clearly in the manuscript.
AUTHOR RESPONSE 3: We do indeed have written consent from the patient whose case is reported in the manuscript. A sentence indicating this has been added to the revised version (lines 87 and 88).
REVIEWER COMMENT 4: Was this investigation approved by the ethics committee? If so, please state clearly in the manuscript and provide details of the ethics committee approbation (code and year).
AUTHOR RESPONSE 4: The research was approved by the Bioethics Committee of our institution. In the revised manuscript, a sentence indicating this approval, along with the code and year of approval by the Bioethics Committee, have been added (lines 88-90).
REVIEWER COMMENT 5: Was there any other analysis done to the patient? HER2? KI67? Even if it is not directly connected to this variant, it still could be of interest for the case analysis.
AUTHOR RESPONSE 5: No other analyses (e.g., HER2, KI67, etc.) have been performed on the patient's sample.
REVIEWER COMMENT 6: Does the patient have a family pedigree with known medical history? Are there any relatives with tumor in the family history? If so, could the authors consider adding it to the manuscript.
AUTHOR RESPONSE 6: We have reviewed all the available information about the patient and have concluded that there is no record of a family history.
REVIEWER COMMENT 7: This manuscript should be revised by the native English speaker as there are some minor errors throughout the manuscript.
AUTHOR RESPONSE 7: The revised manuscript has been reviewed by a native English scientific copy editor.

Reviewer 3 Report
Comments and Suggestions for Authors
The paper submitted for review is a case report of a patient with advanced colorectal cancer (aCRC) in whom next-generation sequencing (NGS) testing revealed a CCDC6::CASP7 gene rearrangement. The discovery of such a genetic rearrangement involved one patient among the 73 consecutives patients with aCRC who were studied.
The paper is professionally written, a case report of the patient is well structured. The description of the new genetic change is documented sufficiently (Figure 1A and B). Other oncogenic mutations (in PIK3CA and KRAS genes) were detected in the patient. In a mature discussion, the authors described the pathophysiological and clinical implications of the presence of the newly described genetic alteration (the first report of CCDC6::CASP7 gene rearrangement), suggesting that demonstration of such an alteration may allow selection of aCRC patients for PARP inhibitor therapy.
I have no substantive comments to the paper. The paper can be published. Congratulations for the authors.
From minor comments,
1. There is no citation of the material included in the supplement of the paper, please check and add if necessary.
2. In the Discussion subsection, the re-explaining of abbreviations, e.g., aCRC (line 98), NGS (line 102), NSCLC (line 105) can be omitted; instead, an explanation of abbreviations used for the first time, e.g., CCDC6 and CASP7 (line 36), PIK3CA and KRAS (line 46), PTEN genes (line 80), and EGFR (line 122) can be added.
Author Response
REVIEWER COMMENT 1:
The paper is professionally written, a case report of the patient is well structured. The description of the new genetic change is documented sufficiently (Figure 1A and B). Other oncogenic mutations (in PIK3CA and KRAS genes) were detected in the patient. In a mature discussion, the authors described the pathophysiological and clinical implications of the presence of the newly described genetic alteration (the first report of CCDC6::CASP7 gene rearrangement), suggesting that demonstration of such an alteration may allow selection of aCRC patients for PARP inhibitor therapy.
I have no substantive comments to the paper. The paper can be published. Congratulations for the authors.
AUTHOR RESPONSE 1: We would like to thank the Reviewer for their congratulations and very positive comments.
REVIEWER COMMENT 2: There is no citation of the material included in the supplement of the paper, please check and add if necessary.
AUTHOR RESPONSE 2: We would like to thank the Reviewer for pointing out this omission. We have now added this citation on page 2, lines 48 and 73.
REVIEWER COMMENT 3: In the Discussion subsection, the re-explaining of abbreviations, e.g., aCRC (line 98), NGS (line 102), NSCLC (line 105) can be omitted; instead, an explanation of abbreviations used for the first time, e.g., CCDC6 and CASP7 (line 36), PIK3CA and KRAS (line 46), PTEN genes (line 80), and EGFR (line 122) can be added.
AUTHOR RESPONSE 3: Following the reviewer’s suggestion, in the revised manuscript, we have omitted the abbreviations “aCRC” (line 106), “NGS” (line 110), and “NSCLC” (line 113) from the Discussion section, and we have included explanations for the abbreviations “CCDC6” and “CASP7” (line 36), “PIK3CA” and “KRAS” (lines 46 and 47), “PTEN” genes (line 83), and “EGFR” (line 130).

Round 2
Reviewer 1 Report
Comments and Suggestions for Authors
After review of the authors' responses, they are inadequate to address the main issue with this report, which is that there are many thousands of fusions identified in cancers, but very few are actually oncogenic. If each one is published as a case report without any functional validation - indeed, without even any bioinformatic prediction of the fusion's function - there would be an endless number of such reports. Without confirmation that this fusion is actually a driver event, the case report is not of value.
Author Response
AUTHOR RESPONSE 1: We acknowledge the reviewer's comments. Different fusion partners of CCDC6 (exon 2) have been identified, with RET and ALK being the most frequent. This demonstrates that the CCDC6 gene has a high susceptibility to chromosomal recombination. The role of CCDC6 (exon 2) in these fusions has been well documented in the literature. Indeed, the fusion of CCDC6 (exon 2) with other genes results in a loss of function of the CCDC6 protein, so we can infer that such loss of function also occurs when CCDC6 is fused with other genes such as CASP7. This is explained in the discussion section of the manuscript.
